# Formulation Development of Meloxicam Binary Ethosomal Hydrogel for Topical Delivery: In Vitro and In Vivo Assessment

**DOI:** 10.3390/pharmaceutics16070898

**Published:** 2024-07-04

**Authors:** Ahdaq Ali Faisal Al-Ameri, Fatima Jalal Al-Gawhari

**Affiliations:** 1Iraqi Ministry of Health, Nasiriya 64001, Iraq; 2Department of Pharmaceutics, College of Pharmacy, University of Baghdad, Baghdad 10071, Iraq

**Keywords:** arthitis, binary ethosome, Box–Behnken, Meloxicam, topical delivery

## Abstract

The article aimed to formulate an MLX binary ethosome hydrogel for topical delivery to escalate MLX solubility, facilitate dermal permeation, avoid systemic adverse events, and compare the permeation flux and efficacy with the classical type. MLX ethosomes were prepared using the hot method according to the Box–Behnken experimental design. The formulation was implemented according to 16 design formulas with four center points. Independent variables were (soya lecithin, ethanol, and propylene glycol concentrations) and dependent variables (vesicle size, dispersity index, encapsulation efficiency, and zeta potential). The design suggested the optimized formula (MLX–Ethos–OF) with the highest desirability to perform the best responses formulated and validated. It demonstrates a 169 nm vesicle size, 0.2 dispersity index, 83.1 EE%, and −42.76 mV good zeta potential. MLX–Ethos–OF shows an amorphous form in PXRD and a high in vitro drug release of >90% over 7 h by diffusion and erosion mechanism. MLX–Ethos–OF hyaluronic acid hydrogel was fabricated and assessed. It shows an elegant physical appearance, shear thinning system rheological behavior, good spreadability, and skin-applicable pH value. The ex vivo permeation profile shows a flux rate of 70.45 μg/cm^2^/h over 12 h. The in vivo anti-inflammatory effect was 53.2% ± 1.3 over 5 h. compared with a 10.42 flux rate and 43% inflammatory inhibition of the classical ethosomal type. The conclusion is that binary ethosome is highly efficient for MLX local delivery rather than classical type.

## 1. Introduction

Arthitis is a joint inflammation derived from the Greek “Arthon”, which refers to joints, and the Latin “Itis”, which specifies inflammation. Inflammatory arthitis is caused by autoimmune processes such as rheumatoid arthitis, osteoarthitis, ankylosing spondylitis, etc. [1,2]. The inflammatory response and pain associated with previous diseases may be inhibited using Non-Steroidal Anti-Inflammatory Drugs (NSAIDs) [3].

Meloxicam (MLX) is a non-selective, non-steroidal anti-inflammatory drug with a higher selectivity for cyclooxygenase COX2 (inducible isoform) than cyclooxygenase COX1 enzyme (constitutive isoform) [4]. MLX belongs to class IIa in the pharmaceutical classification system PCS because of pKa values 1.1 and 4.2 [5]. In 2000, the United States Food and Drug Administration (US FDA) approved MLX for clinical use in treating skeletal muscle pain and arthitis-related inflammation [6,7]. However, MLX long-term use at high doses can lead to systemic adverse events such as gastrointestinal GI ulcers/bleeding [8,9,10] and hepatic, nephotic, and cardiovascular toxicity [11,12]. Therefore, MLX is a potential candidate for topical delivery, especially for patients suffering from peptic ulcers or GI problems. Topical delivery, a highly efficient method for drug delivery to a precise site, offers a superior alternative to oral administration in many cases. The local delivery of MLX to a specific muscle or joint is not only helpful in avoiding first-pass metabolism but also in preventing side effects such as gastrointestinal upset or pH-dependent solubility. Moreover, it enhances patient compliance [13,14].

Unfortunately, MLX topical delivery faces challenges such as meloxicam solubility and permeability though the skin’s outer layer stratum corneum (SC), which provides a rich research area to carry the drug inside. Liposomes, introduced in 1980 by Mezei & Gulasekharam, were a significant step in drug delivery across the skin. However, the challenges associated with vesicle rigidity and a high drug percentage deposited on the skin. In 2000, Ethosomes, advanced liposomal derivative carriers introduced by Touitou, consisted of a phospholipid (PL), a high ethanol content of 50 *w*/*w*%, and water [15,16,17] that exhibited more flexibility and deformability than liposomes because ethanol presence that increased the lipid fluidity. Ultra-deformable ethosomes, a recent development, are binary ethosomes composed of PL, ethanol, and another alcohol, mostly propylene glycol PG or isopropyl alcohol IPA [18,19,20,21,22].

The research includes two alcohol types: ethanol, which represents the backbone for the ethosomal system, and PG, which provides better solvency to MLX than IPA. Two alcohols produce a synergism (additional) effect on the MLX moiety to escalate its solubility through the system and solubilize the ethosomal lipid, further reducing the vesicular size. Lipid solubilization facilitates MLX vesicular entry and increases encapsulation efficiency. Furthermore, alcohols negatively affect zeta measurement by increasing the negative charge presence around the vesicles. Therefore, they enhance the system stability and vesicle-skin interaction that promotes high permeability [23,24,25], providing a novel, promising approach for topical delivery.

Furthermore, binary ethosome provides rapid drug delivery and release over other vesicular systems, such as liquid crystalline cubosome and hexosome, especially for hydrophobic moieties such as MLX. Because it is difficult to escape from the vesicular system when incorporated inside the cubosome or hexosome, thus providing sustained drug release [26]. The rapid relief of pain and inflammatory response was intended from the topical use. However, Wu Y et al. successfully delivered intranasal plasminogen-based nanomedicine (hydrophobic moieties) through hexosome for the Parkinson’s mouse model [27]. Previous studies have also aimed to solve the MLX permeation issue, such as classical ethosomes [28] that significantly enhance drug delivery and anti-inflammatory response. This research compares the difference between two ethosomal types besides enhancing MLX solubility and delivery.

The research aimed to develop an MLX-binary ethosome as an innovative method that can potentially revolutionize drug delivery, escalating MLX solubility for topical application by avoiding systemic adverse events, enabling deep drug penetration into skin layers, and enhancing patient compliance. This method paves the way for a new pharmaceutical era by comparing the results with the classical ethosomal type. The ethosomal dispersion was incorporated into the hyaluronic acid hydrogel. Hyaluronic acid is a natural glycosaminoglycan polysaccharide that represents the main skin component responsible for a skin structure. Hydrogel was investigated for its rheologic and thixotropic behavior by viscosity measurement, spreadability upon application, pH measurement, ex-vivo permeation, and in vivo anti-inflammatory effect of MLX on rat-induced paw edema.

## 2. Materials and Methods

### 2.1. Materials

The hot method was used in the formulation. Meloxicam yellow powder, purity 99.0%, CAS:71125-38-7 was purchased from Meyer (Shanghai) Biochemical Technology Co., Ltd., Shanghai, China. Soya bean lecithin HPLC 98% lyophilized, white to yellow to brown powder, Molecular Weight 758.06 g/mol., CAS:8002-43-5 was purchased from Suzhou Nuopal New Material Technology Co., Ltd., Suzhou, China. Honeywell, Böblingen, Germany, manufactured absolute ethanol 99%. Thomas Baker, Mumbai, India, manufactured propylene glycol. Phosphate buffer saline pH 7.4 was manufactured by Hi. Media Laboratories, Mumbai, India. Al Basheer Scientific Bureau, Basera, Iraq, manufactured deionized water. Hyaluronic acid was purchased from Hyperchem, Hangzhou, China. The Iraqi Ministry of Higher Education and Scientific Research ethical committee, Baghdad University, College of Pharmacy (approval number: RECAUBCP472023K) approved the animal model used in this research.

### 2.2. Methods

#### 2.2.1. Formulation (Optimization) by Box–Behnken Design Expert

The Box–Behnken design expert software version 13.0.5, state ease, USA, was used for factors optimization of MLX binary ethosomes to produce the best responses. MLX-binary ethosome preparation was designed using three factors and four center points. The independent variables (factors) were the soya lecithin (2–4% concentration), ethanol (15–45%), and propylene glycol PG (0–20%), and the dependent variables (responses) which are vesicle size (nm), dispersity Index (unit less), encapsulation efficiency %, and zeta potential (mV). The response criteria were minimization of vesicle size and dispersity index, targeting the highest encapsulation efficiency value and the zeta potential within its range. All responses have the same importance. The general equation was given as follows:Yi = b0b1X1 + b2X2 + b3X3 + b12X1X2 + b13X1X3 + b23X2X3 + b11X12 + b22X22 + b33X32(1)
where: Yi dependent variables, b0 intercept, b1,2,3 regression coefficients, and Xi independent variables.

#### 2.2.2. MLX-Binary Ethosomes Preparation Method

MLX-binary ethosomes formulas were prepared using the hot method [23] according to experimental runs suggested by the Box–Behnken. The phases were prepared separately in two glass tubes using a 7 mm × 15 mm cylindrical magnetic bar at 40 °C. The organic phase consisted of MLX, ethanol, and PG, and the aqueous one contained soya lecithin dispersion in deionized water. The organic phase dripped to the aqueous using a syringe pump at 200 μL/min rate through a 23-gauge needle with continuous stirring using a magnetic stirrer/hotplate (witeg Labortechnik GmbH, Seoul, Korea distribution partner) at 500 rpm for one hour. After that, the formula rested 15 min before further vesicle size reduction by a probe sonicator (500 Watt, 20 kHz, Qsonica, 53 Church Hill RD, Newtown, CT, USA) at a frequency of 30% amplitude, 2 s on, 2 s off for half minutes. The preparations were stored in a dark container at 4 °C for evaluation (Figure 1).

#### 2.2.3. MLX-Binary Ethosomes Validation

##### Vesicle Size and Dispersity Index Measurements

The vesicle size was estimated using dynamic light scattering DLS at 173° Ѳ (Ultra-red Zeta sizer, Enigma Business Park Grovewood Rd, Malvern, UK). Deionized water was used to dilute the sample in a ratio of 1:20 *v*/*v* to ensure proper light scattering by the Brownian motion of the vesicle. The dispersity index measures homogeneity or size distribution in a sample. In pharmaceutics lipid carrier delivery systems, the dispersity is preferred to be less than 0.3, indicating homogenous preparation [29,30,31]. All readings were performed in triplicate and expressed as Mean ± Standard Deviation (M ± SD, *n* = 3).

##### Zeta Potential Measurement

The zeta sizer measures the charge type and intensity based on electric mobility. It measures the charge between the particle surface and the slipping plane. The sample was diluted up to 1:20 *v*/*v* with deionized water. The procedure was performed in triplicate.

##### Encapsulation Efficiency % Calculation

Encapsulation efficiency % is the MLX amount entrapped into vesicles of preparation. The indirect method was employed by an ultra-centrifugal filter technique (4 mL, NMWCO 10 kDa, Millipore, Merck, Darmstadt, Germany). The sample was centrifuged using a cooling centrifuge at 4 °C, 12,000 rpm for 30 min. After that, the withdrawn supernatant was diluted with ethanol and measured by a UV spectrophotometer at ƛ_max_ 365 nm. The unencapsulated MLX amount was estimated from the calibration curve equation of MLX in ethanol (slope 0.0466, intercept +0.007, R^2^ 0.9999) and subtracted from the total drug content in the formulation (drug content of all formulation 10 ± 2 mg) divided by the total drug content. The equation calculated encapsulation efficiency is as follows:EE% = ((T − UE)/T) × 100(2)
where T is the total amount of MLX content in the formulation, and UE is the unencapsulated MLX amount.

#### 2.2.4. MLX-Binary Ethosomes Optimized Formula (MLX–Ethos–OF) Suggested by Box–Behnken

The main objective of the Box−Behnken design of the experiment was to obtain an optimized formula as a solution with the highest desirability (0.933) to obtain the best response, smaller vesicle size and DI, highest EE%, and within-range zeta potential. The optimized formula was prepared using the same method (hot method) using 3% soya lecithin, 25.9% ethanol, and 4.9% PG. The response was evaluated, and the relative error % [32,33,34] was estimated according to the equation:Relative Error % = ((predicted value − actual value)/predicted value) × 100(3)

Relative error percentage was used to measure the error percent in MLX–Ethos–OF evaluation measurements. The formula was stored in a refrigerator at 4 °C.

#### 2.2.5. Evaluation of MLX–Ethos–OF

##### Vesicle Size, Dispersity Index, EE%, and Zeta Potential Validation

MLX–Ethos–OF evaluation was performed using a similar technique and apparatus used for MLX-Ethos evaluation.

##### PXRD Analysis of MLX–Ethos–OF

MLX–Ethos–OF was dried for 48 h. using a (TELSTAR freeze dryer device under 0.700 mBar vacuum pressure, 316 L stainless steel condenser at −55 °C temperature, Barcelona, Spain). The obtained powder was used for powder X-ray diffraction (PXRD) analysis using (AERIS XRD Diffractometer, Malvern, Almelo, The Netherlands) to evaluate the formation of MLX amorphous form.

##### MLX–Ethos–OF In Vitro Release Study

MLX release study from MLX–Ethos–OF used a dialysis membrane (8000–14,000 MWCO) in a type II dissolution apparatus (RC-6 Dissolution tester, Faithful, Hebei, China). The release media was 500 mL of phosphate buffer saline PBS pH 7.4 at 37 ± 0.1 °C rotated at 100 rpm. The sink condition was maintained during the process.

The sink condition was obtained when the dissolution media was at least three times the volume of the dissolved MLX dose. It was measured depending on MLX saturated solubility in PBS pH 7.4, which measured 0.41 mg/mL [35]. Three milliliters were sampled and replaced directly by PBS 7.4 at 5, 15, 30, 60, 120, 180, 240, 300, 360, and 420 min. The amount of MLX released was measured using a UV spectrophotometer (6100 PC double beam, EMC LAB, Berlin, Germany) at λ_max_ 362 nm. MLX release % was estimated using its calibration curve equation of MLX in PBS pH 7.4 (slope 0.0496, intercept +0.0027, R^2^ = 0.9998). The release was performed in triplicate. Microsoft Excel 2016 D.D. Solver^®^ was used to measure the similarity factor (f2) between MLX–Ethos–OF and MLX–PBS pH7.4 suspensions as a reference (MLX is a weakly acidic drug that poses a pH-dependent solubility and is expected to have higher solubility than pure MLX in water). If f2 < 50, indicating a significant difference between the tested and reference formula. Otherwise, the profiles are considered similar.

##### Release Kinetic Models

D.D. solver software added in Microsoft Excel 2016 was used to study the release kinetic models and release mechanisms using the drug release data. Using the dissolution data were fitted with Zero, First, Higuchi, Korsmeyer–Peppas, and Baker–Lonsdale models. The equation of each model is described as the amount of MLX released at a time with a specific release constant K for each model. The n value is the MLX release exponential that characterizes the drug release mechanism. The best-fitting model was chosen depending on the five models’ highest R square, lowest AIC, and high MSC values [36].

The zero-order model equation is:Q_t_ = Q_0_ + k_0_(4)
where Q_t_ is the drug released after time t, Q_0_ is the initial drug amount, and K_0_ is the zero-order rate constant.

The first-order model equation is:Log Q_t_ = Log Q_0_ + K_1_t/2.303(5)
where, Q_t_ is the drug released after time t, Q_0_ is the initial drug amount, and K_1_ is the first-order rate constant.

The Higuchi model equation is:Q = k_H_√t(6)
where, Q is the amount of released drug after time t, and K_H_ is the Higuchi constant.

The Korsmeyer–Peppas model equation is:Log Mt/M∞ = log k_KP_ + *n* log(7)
where, Mt is the amount of drug released at time t, M∞ is the cumulative amount of drug released at infinite time, K_KP_ is the Korsmeyer–Peppas constant, and *n* is the release exponent; its value characterizes the mechanism release. The *n*-value Interpretations are when *n* < 0.5 quasi Fickian diffusion transport mechanism and non-swellable matrix diffusion release mechanism, *n* = 0.5 Fickian diffusion transport and the same quazi type release mechanism, 0.5 < *n* < 1 anomalous transport and both diffusion and relaxation (erosion), *n* = 1 case II transport zero-order release, *n* > 1 super case II transport relaxation/erosion.

The Baker–Lonsdale model equation is:f = 3/2 [1 − (1 − Mt/Mα)^2/3^] − Mt/Mα = K_t_(8)
where Mt/Mα is the fraction of the drug released at time t, and K_t_ is the release constant. Baker–Lonsdale is used to describe drug release from sphere particles.

##### MLX–Ethos–OF Imaging by Transmission Electron Microscopy (TEM)

JEOL JEM-2100F transmission electron microscopy was used for sectional visualization of MLX–Ethos–OF. A drop of ethosomal formulation was diluted and dried on a carbon-coated grid at room temperature, 25 °C. After that, the sample was stained with 2% uranyl acetate for 5 min before being viewed under the microscope with 100 k magnitude and an accelerating voltage of 80 kV.

#### 2.2.6. MLX–Ethos–OF Hydrogel Preparation and Validation

##### MLX–Ethos–OF Hydrogel Preparation

Hydrogels are a 3D network of cross-linked natural or synthetic polymers that are easily applicable and most compliant with the semisolid formulations of the patient. MLX–Ethos–OF was amalgamated into 1% *w*/*v* and 2% *w*/*v* Hyaluronic acid H.A. gel base by the cold method [37]. Briefly, the accurate weight of H.A. was dispersed in water and stirred at 2000 rpm, at 20 °C for 2 h. The dispersion was stored overnight in the refrigerator for hydration assurance. After that, MLX–Ethos–OF amalgamated into the H.A. homogeneous dispersion in a 1:1 ratio and mixed with a spatula until the gel became homogenous, soft, and consistent. Two drops of peppermint oil were added to overcome the ethanoic smell. Then, the volume was completed with glycerin to the final hydrogel weight. The same technique was used to prepare plain MLX hydrogel.

##### MLX–Ethos–OF Hydrogel Validation

Physical Appearance

Visual and sensory validation includes evaluating the hydrogel’s apparent color, softness, homogeneity, and consistency.

Hydrogel Viscosity

The viscosity of MLX–Ethos–OF hydrogel was evaluated by the viscometer (Myer rotary viscometer, Vendrell instrument, El Vendrell, Spain). Using spindle number 4 at room temperature, different rpm (shear stress) was used to evaluate different polymer concentrations’ rheologic behavior [38]. The thixotropic behavior was examined when stress decreased. The viscosity test was performed in triplicate.

Hydrogel pH Measurement

Human skin is acidic, plays a role in body protection, and avoids stratum corneum SC integrity destruction [39,40]. Skin pH varies due to numerous factors such as disease, detergent, age, and sweat components [41]. Therefore, the skin pH has essential properties that must validated to ensure formulation safety and reduce irritation. 1 mL of the hydrogel was applied in 10 mL of deionized water and sonicated for 5 min [42]. The pH was read using a pH tester (Hanna, Nușfalau, Romania). The technique was done in triplicate.

Spreadability Test by Parallel Plates

Spreadability is a feature that explains how hydrogel can spread quickly upon application. The area of the applied gel was measured by applying 0.5 mL of gel on a glass plate using a sterile syringe and covering it with a corresponding plate in a precise position. A 0.5 kg weight was applied to the glass plate to calculate the area of the spreading hydrogel circles. The difference in circle area (cm^2^) before and after the force (weight) applied was measured [43]. The spreadability test was performed in triplicate as follows:A = πr^2^(9)
where A is the hydrogel spread area and r is the cycle radius after weight application

#### 2.2.7. Ex Vivo Permeation of MLX–Ethos–OF Hydrogel

Permeation study was performed in a vertical Franz diffusion cell through a hairless rat skin [44]. Firstly, a Wistar albino rat was anesthetized with 10% ketamine and xylazine HCL before being sacrificed, and the skin hair was removed using a hair removal cream. Secondly, the skin was separated from the subcutaneous tissue, washed with PBS pH 7.4, and stored in PBS pH 7.4 at −20 °C until use within 24 h. The franz effective diffusion area was 1.7 cm^2^. The skin was fixed between the donor and receiver parts (chambers). The receiver part capacity was 12 mL filled with PBS pH 7.4. The system was heated to 37 °C and magnetically stirred at 200 rpm.

After that, 1g of MLX–Ethos–OF hydrogel was placed on the skin in the donor part and covered with cellophane foil. At predetermined times, 1 mL was sampled from the receiver compartment and replaced with PBS 7.4 to ensure the sink condition. The amount of MLX was estimated by a UV spectrophotometer. The procedure was performed in triplicate and the cumulative amount of MLX permeated was calculated as follows:The cumulative MLX amount = V_R_ × C_n_ + [V_S_ (∑ C_1_ + C_2_ +…+ C_n−1_)](10)
where the receptor volume is V_R_, the volume of withdrawal samples each time is V_S,_ and C_n_ is the sample concentration at the nth time. The calculated cumulative amount was then divided by the Franz surface area in cm^2^ to obtain the permeation in μg/cm^2^.

The steady–state flux (J_SS_) was calculated from the slope after plotting cumulative MLX permeated per unit area and time (μg/cm^2^/h). The permeability coefficient (K_p_) was mathematically calculated by dividing the flux by the initial MLX concentration in MLX–Ethos–OF in the donor compartment [33]. The statistical analysis was performed in Graph Pad Prism 8.0.1 using an unpaired t-test as follows:K_p_ = J_ss_/C_0_(11)

#### 2.2.8. In Vivo Anti-inflammatory Effect of MLX–Ethos–OF Hydrogel

Six healthy male Wistar albino rats were used for an in vivo anti-inflammatory effect study of MLX–Ethos–OF hydrogel in egg albumin-induced paw edema in the rat’s right hand [45]. The rats were fed a dried mixture of seeds, grains, and vitamins and drank tap water. Egg albumin was injected into the plantar surface of the right hand, and edema formed after 30 min. Vernier caliper was used to measure edema size. The rats weighed 300 ± 50 g and were divided into three groups randomly. The first group was the control group without medication, the second group was treated with 0.5 g MLX–Plain hydrogel, and the third group was treated with 0.5 g MLX–Ethos–OF hydrogel and waited for the hydrogel to dry on the skin of the hand. The edema size was measured 5 h after the treatment, and the percentage of inflammatory inhibition was estimated according to the following equation:Inhibition of edema % = S_0_ − S_t_/S_0_(12)
where the treatment group’s edema size is St, and S_0_ is the control group.

## 3. Results and Discussion

### 3.1. Box-Behnken Design Expert Optimization Results

The Box–Behnken design expert was used for optimizing MLX-binary ethosome variables, statistical analysis, obtaining a general equation of the model, 3D surface response, and contour graphs for each dependent factor (response) [46,47]. The design equations for each response are presented in the Appendix A. The results of the experimental work are illustrated in Table 1.

The regression coefficient indicates the model suitability validation [48]. The R^2^ value ranged from 0.70 to 0.99, indicating a significant way in which responses followed the independent factors, not by chance [49]. Response regression analysis is explained in Table 2. The coefficient of variation (C.V) is a standard deviation SD to the mean M ratio, which measures the distribution out of the mean. Adequate precision is the signal–to–noise ratio of each response, which is preferred to be higher than 4, elucidating the signal area measure [50]. Response regression analysis is explained in Table 2.

The Box–Behnken design expert is used to obtain a precise statistical analysis. ANOVA gives an idea about the significance of the model fit the response, where the lack of fit must not be significant. Also, the significance of independent variables affects the dependent one with a *p*-value < 0.05 at a 95% assurance level. All responses fit the model well except for the dispersity that significantly lacks fit and thus gives an imprecise reading. Each dependent variable is affected by a specific factor(s). ANOVA Analysis is shown in Table 3.

### 3.2. Independent Variables Impact Responses

#### 3.2.1. Independent Variables Impact the Vesicle Size

The vesicle size response fitted the quadratic model, ranging 107 ± 3.99 to 534 ± 74.55. From Table 1, the lipid affects the vesicle size by increasing its concentration from 2 to 4% in formula or (runs number) 1 and 14, increasing the vesicle size from 174 to 288 nm, which also impacts the dispersity by increasing the dispersity value thus reported in the literature [51]. Ethanol and PG significantly impact the vesicle size (*p*-value < 0.0001) due to their solubilizing effect on the phospholipid [52,53]. The regression analysis model reported a high R^2^ of 0.9975, referring to the independent variable significantly impacting the vesicle size with <0.2 difference between predicted and actual readings at a high adequate precision of 95.22, indicating the measurement in the signal area. Figure 2 shows a 3D surface plot of the variable’s effect on the responses where the red balls represent the point of the best response value. Figure 3 shows the contour graph for each response where red represents the high value, while blue represents the low value. The factors’ interaction on responses can illustrated in Figure 4, where VS a and b straight or curved lines (not crossed) indicate no interaction, while crossed lines in VS c refer to a significant effect of ethanol and PG on vesicle size by additional solubilizing effect.

#### 3.2.2. Independent Variables Impact Dispersity Index

The dispersity index fitted the quadratic model with values ranging 0.17 ± 0.017 to 0.60 ± 0.263. From Table 1, run numbers 8 and 12 demonstrate no significant dispersal increase with soya lecithin concentration increase (*p*-value 0.62). Run numbers 11 and 12 show that ethanol significantly impacts dispersity (*p*-value 0.007) by lipid solubilizing effect increasing dispersity, while PG produces insignificant dispersity minimization in runs 13 and 14. The moderate regression coefficient R^2^ was 0.7 at 7.9 adequate precision, and the model has a significant lack of fit; therefore, the model is not suitable for dispersity prediction (the fitting is necessary), and the negative predicted value indicates that the mean is the best predictor for the dispersity index [54]. Factor interaction was reported that all variables affect the dispersity index substantially.

#### 3.2.3. Independent Variables Impact the Encapsulation Efficiency %

The response fitted the quadratic model and ranged 14 ± 0.08 to 88 ± 0.02. From Table 1, run numbers 4 and 6 show that the negative effects of ethanol concentrations at 3% soya lecithin beyond a specific level led to a decrease in drug entrapment because of vesicle destruction or leak. 13 and 14 runs have 4% soya lecithin, and an increase in PG percent from 0 to 20% leads to sufficient solubilization for lipid content and increased encapsulation efficiency [51,55,56]. In contrast to most popular research, an increase in lipid concentration leads to a significant decrease in drug entrapment, as in formulas 1 and 14. The main cause is unclear, but it may be because of the lower drug–to–lipid ratio where no sufficient drug concentration is to be entrapped or because of the occlusion of ultra-centrifugal filter pores. There is considerable interaction between soya lecithin and ethanol, soya lecithin and propylene glycol, and ethanol and propylene glycol on encapsulation capacity.

#### 3.2.4. Independent Variables Impact Zeta Potential

All ethosomal vesicles have a negative charge layer surrounding them, providing a repulsion force that enhances their stability. The measured zeta potential ranged −36.3 ± 3.11 to −51 ± 0.52. All variables (soya lecithin, ethanol, PG) have a positive impact, indicating that all factors negatively affect zeta measurement. Ethanol and propylene glycol have a significant interaction that provides excess negativity (synergism) around the vesicles; therefore, the zeta potential fits the 2FI model (two factorial interactions) [57,58]. The soya lecithin percentage used in the study did not significantly affect the zeta potential, possibly because of the ethanol solubilizing effect [59]. The result was measured with a high adequate precision of 14.98. A significant interaction between soya lecithin and ethanol on zeta potential. These interactions made a severe need for variable optimization to develop a binary ethosome with the best response criteria.

### 3.3. MLX–Ethos–OF Validation

#### 3.3.1. Vesicle Size, Dispersity Index, Encapsulation Efficiency %, and Zeta Potential of MLX–Ethos–OF

MLX–Ethos–OF vesicle size of 169 nm, which is <200 nm and suitable for skin delivery [60,61], dispersity 0.20 indicates homogenous preparation, high encapsulation efficiency of 83.1%, and zeta potential of −42.76 mV indicates good ethosomal suspension stabilization. Figure 5 shows the measurement peaks.

Relative error is a measure of the error percent in MLX–Ethos–OF formulation. Relative error % is the highest in the dispersity index and thus may happen because of the model’s lack of fit and the lowest value in zeta potential. Relative error % is shows in Table 4 and is preferred to be <10%.

#### 3.3.2. PXRD Analysis

The powder X-ray diffraction graph of pure MLX shows a distinctive peak at 11.2°, 13.38°, 14.75°, 14.89°, 15.84°, 17.79°, 18.48°, 19.17° indicating crystal form peaks similar to the published research [62]. The absence of all sharp, highly intense peaks of pure MLX and the presence of broad halos between 10 and 30 Ѳ in MLX–Ethos–OF indicate an amorphous state in lyophilized MLX–Ethos–OF. It is lacks the ordered shape and is more soluble than the crystal form, indicating increased MLX solubility in the ethosomal system. Figure 6 demonstrate the PXRD peaks of MLX–Ethos–OF and pure MLX.

#### 3.3.3. In Vitro Drug Release% from MLX–Ethos–OF Nanosuspension

The in vitro drug release of MLX–Ethos–OF reached an average release of 90.5% ± 4.29 after 7 h, and MLX–PBS reached 53.5% ± 2.2 simultaneously, as shown in Figure 7. An increase in the solubility of MLX–Ethos–OF due to complete amorphous form formation leads to higher drug leakage from the vesicles though the dialysis membrane. D.D. solver similarity test reported that the observed f2 was 23.48, indicating a significant difference between the reference and tested dissolution profile (f2 < 50 indicated different profiles).

#### 3.3.4. Release Kinetic Models’ Fitness

The release data were fitted with five kinetic models using a D.D. solver in Microsoft Excel 2016. The selection of a more suitable model depends on several parameters such as R^2^ adjusted, the AIC, and MSC. The adjusted R^2^ is the determination coefficient used to evaluate the best–fitted model, where the highest value indicates the best one [63,64,65]. The AIC (Akaike Information Criteria) is used for model selection where the lower value indicates high fitting and model suitability. Also, the high–value MSC (MicroMath Corporation) was used to select a suitable model [66].

The results of release kinetics fitted models are shown in Table 5. Korsmeyer–Peppas model demonstrates the highest R^2^, lower AIC, and high MSC; therefore, it is suitable. An n-value of 0.799 describes MLX anomalous transport (non-Fickian), and the release mechanism was diffusion and erosion from MLX–Ethos–OF.

#### 3.3.5. TEM Imaging

The ethosomal suspension was magnified at 100× to visualize the vesicular system. TEM shows a spherical to oval dark vesicle with a uniform appearance and a size within the range of the DLS measure of the zeta sizer. TEM size ranges from 100–200 nm and is thus reported in the literature [67] to be approximately the same size, which indicates homogeneity. Figure 8 shows a sectional image of the ethosomal vesicles using TEM.

#### 3.3.6. Hydrogel Validation

##### Hydrogel Appearance

Hydrogel visually appears as a thick, consistent dispersion with a light pastel yellow color. It is stable and homogenous, has no aggregation (soft, uniform texture), is not greasy, and has a peppermint oil odor.

##### Hydrogel Viscosity

When the rpm of the viscometer (shear stress) increases, the internal structure of the hydrogel is destroyed (viscosity decreased) and starts flowing; therefore, the hydrogel shows shear-thinning rheological properties. The hydrogel viscosity decreases when shear stress increases and returns to its origin when the stress is removed. This behavior is called thixotropic behavior, a desirable property in pharmaceutics that impacts the manufacturing process and dose administration (hydrogel extrusion). Furthermore, the viscosity increases as polymer concentration increases from 1% to 2% at the same rpm due to more intermolecular entanglements with high polymer concentration. The viscosity results are shown in Figure 9.

##### pH and Spreadability Measurement

All prepared hydrogels have a pH ranging from 6 ± 0.1 to 6.3 ± 0.3, which is near the skin pH, does not induce irritation or itching upon application, and is accepted in the literature [68,69]. Hydrogel spreadability refers to the extent to which the hydrogel spreads on the skin upon application. The hydrogel spreadability decreases as the polymer concentration increases. Table 6 illustrate the pH and spreadability measurements. Table 6 shows hydrogels pH and spreadability measurememts.

##### Ex Vivo Permeation of MLX–Ethos–OF Loaded Hydrogel

From Figure 10, effective permeation of MLX–Ethos–OF with flux representing the slope of the equation straight line (linear portion) 70.45 μg/cm^2^/h was observed. Binary ethosome has flexibility and deterioration character though stratum corneum skin layer ethanol and propylene glycol presence that solubilizes the skin lipid layer and delivers MLX inside. In contrast, MLX–Plain hydrogel has a flux rate of 0.6812 μg/cm^2^/h during the same time. The permeation-enhanced ratio was 103, calculated from the following equation [70]:PE = Jss (MLX–Ethos–OF Hydrogel)/Jss of MLX plain Hydrogel(13)
where, PE is the ratio of permeation enhancement, and Jss is the ethosomal and plain hydrogel flux. The permeability coefficient K_p_ was 246.575. Ahad et al. 2014, formulated MLX in the classical ethosomes type. Compared with classical ethosome [28], binary ethosome enhances MLX permeation though the rat skin 7-fold higher than the classical form. The flux rate is 70.45 μg/cm^2^/h for binary ethosome and 10.42 for classical ethosome. The leading cause was the presence of propylene glycol, in addition to ethanol, which enhances ethosome flexibility and penetration in deep layers of the skin by fluidizing the lipid bilayers. Also, they have a synergistic effect with increasing zeta negativity, which aids in vesicle-skin interaction and enhances system stability.

#### 3.3.7. In Vivo Anti-Inflammatory Activity of MLX–Ethos–OF Hydrogel

The Iraqi Ministry of Higher Education and Scientific Research ethical committee, Baghdad University, College of Pharmacy (approval number: RECAUBCP472023K) approved the animal model used in this research. The rats-induced albumin paw edema was the model animal used to study MLX anti-inflammatory response upon topical delivery. Albumin induced acute inflammation and swelling began after 30 min and continued for 5 h of experiments. Figure 11 shows that MLX–Ethos–OF Hydrogel was significantly (*** indicated a *p*-value < 0.001) inhibiting the inflammatory response 53% ± 1.3 during 5 h. In contrast, MLX–Plain hydrogel inhibition was 14.7 ± 0.66. The results were compared with the classical type formulated by Ahad et al. [28], which reported 43% inhibition after 5 h. The leading cause was the ethanol and PG in the binary ethosome, which provided vesicle deformability across the stratum corneum. During the experiment, no rats were irritated or red or had spots upon hydrogel application.

## 4. Conclusions

According to the research results, MLX was successfully formulated as a binary ethosomal hydrogel for topical delivery, introducing a promising approach that increases drug skin permeability and efficacy over the classical type. MLX–Ethos–OF was optimized using the Box–Behnken design, and the optimized formula has a small vesicle size, high entrapment, and the soluble amorphous form of MLX moiety that runs a high in vitro release % >90 over 7 h. It also demonstrates good zeta potential and has a low dispersity index, indicating a stable preparation. MLX–Ethos–OF was incorporated in a hyaluronic acid hydrogel to increase the stability and provide easy skin application. The hydrogel validation shows a shear thinning system with a spreadable and skin-applicable pH value. The MLX–Ethos–OF hydrogel produces a 70.45 μg/cm^2^/h flux rate with a 103 enhancement ratio over MLX–Plainhydrogel and a high permeation rate compared with the classical-type flux rate of 10.42 μg/cm^2^/h. Furthermore, MLX–Ethos–OF produced anti-inflammatory activity of 53.2% ± 1.3 over 5 h of the medication without sensitivity or skin irritation compared with a 43% inhibition rate with the classical type. In contrast, MLX–Plain hydrogel was 14.7 ± 0.66 inflammatory inhibition. However, more research in the future for hydrogel safety and efficacy on human volunteers is necessary after investigating large-scale characterizations of MLX–Ethos–OF hydrogel.

## Figures and Tables

**Figure 1 pharmaceutics-16-00898-f001:**
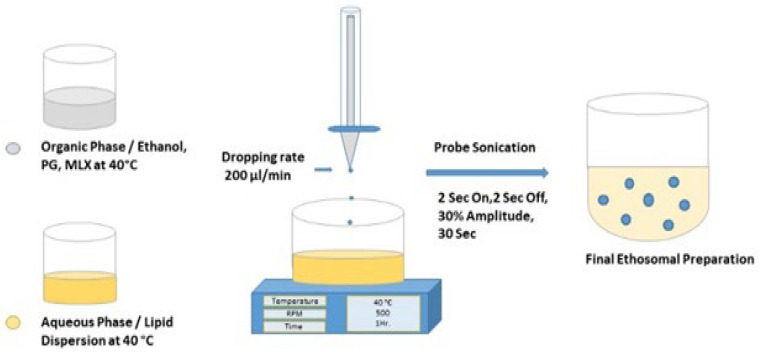
The hot method for MLX-binary ethosome preparation.

**Figure 2 pharmaceutics-16-00898-f002:**
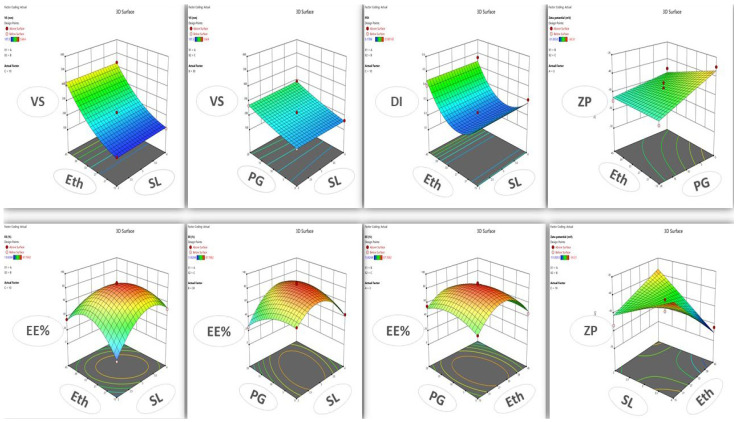
The figure shows a 3D-surface graph of the independent variable’s effect on response. Where the responses (VS is the vesicle size, DI is the dispersity index, ZP is zeta potential, EE% is the encapsulation efficiency), and the variables are (Eth is ethanol, SL is soya lecithin, PG is propylene glycol).

**Figure 3 pharmaceutics-16-00898-f003:**
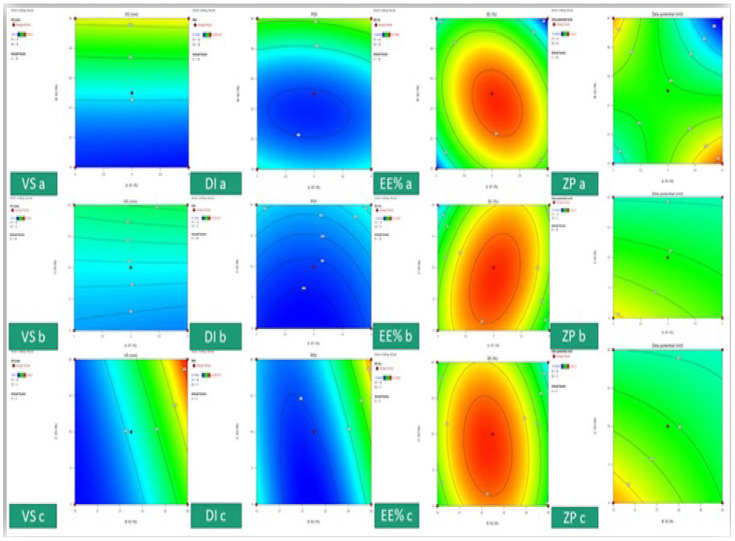
The response contour graph, where the responses are vesicle size (VS), dispersity index (DI), encapsulation efficiency % (EE), zeta potential (ZP), and the variables are represented as a,b, and c where a is the soya lecithin and ethanol effect, b is the soya lecithin effect and propylene glycol, and c is the effect of ethanol and propylene.

**Figure 4 pharmaceutics-16-00898-f004:**
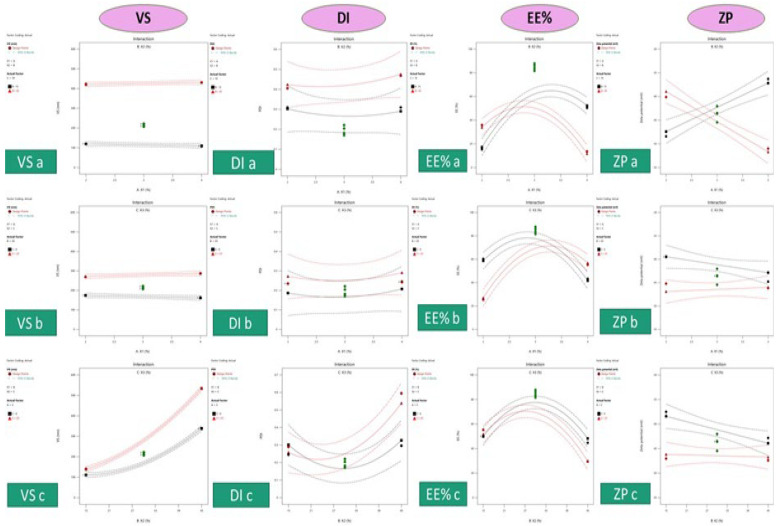
Interaction graphs of each response (where a is the soya lecithin and ethanol interaction, b is the soya lecithin and propylene glycol interaction, and c is the ethanol and propylene glycol interaction.

**Figure 5 pharmaceutics-16-00898-f005:**
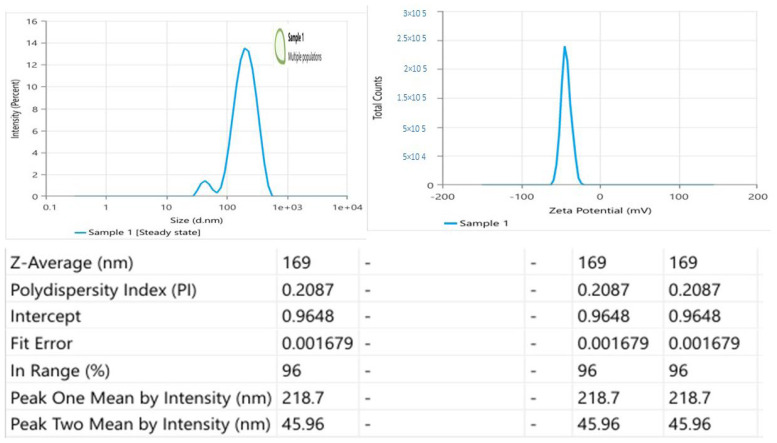
Zetasizer measurement peaks.

**Figure 6 pharmaceutics-16-00898-f006:**
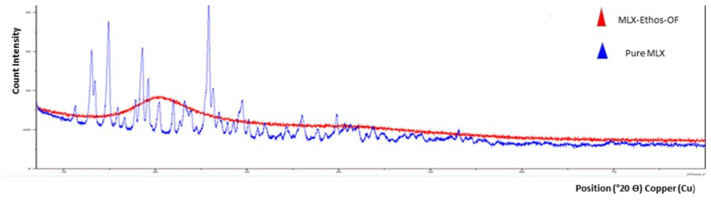
PXRD graph of MLX–Ethos–OF.

**Figure 7 pharmaceutics-16-00898-f007:**
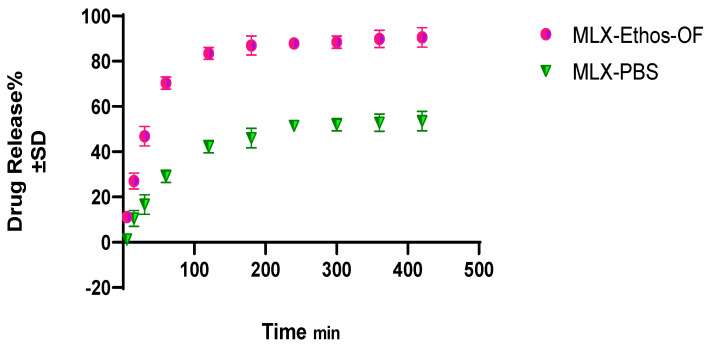
In vitro drug release study of MLX–Ethos–OF and MLX–PBS.

**Figure 8 pharmaceutics-16-00898-f008:**
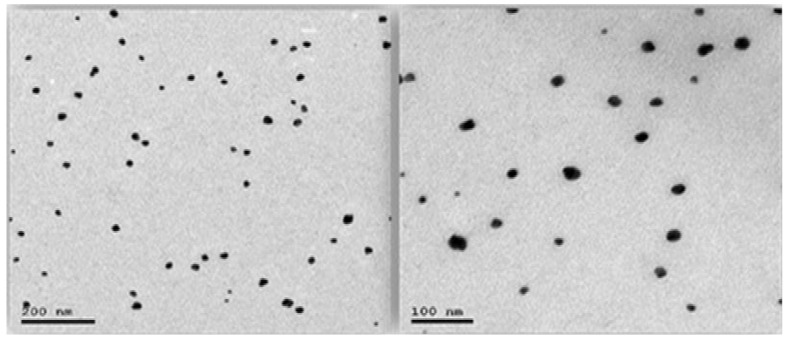
TEM imaging of MLX–Ethos–OF.

**Figure 9 pharmaceutics-16-00898-f009:**
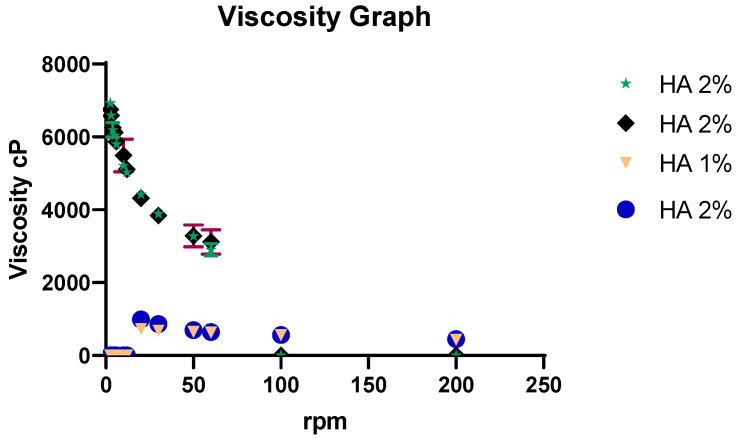
Viscosity graph of hyaluronic acid hydrogel.

**Figure 10 pharmaceutics-16-00898-f010:**
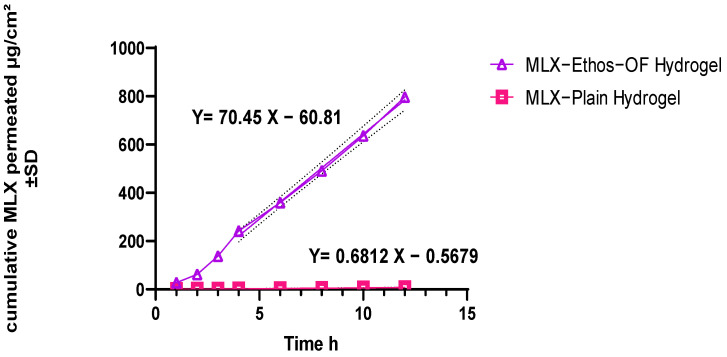
MLX–Ethos–OF and plain MLX hydrogel permeation.

**Figure 11 pharmaceutics-16-00898-f011:**
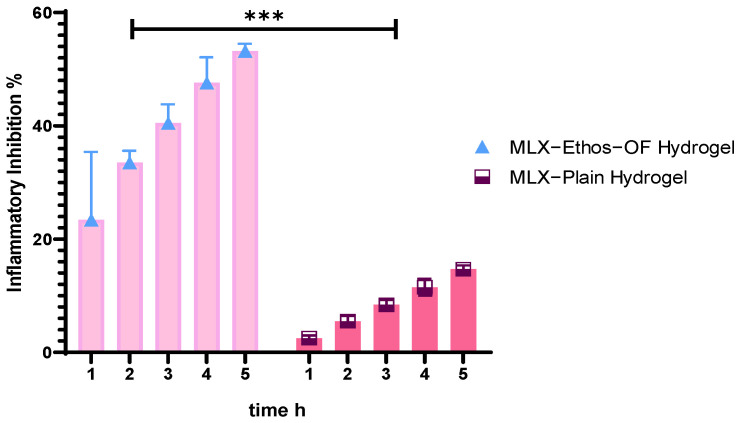
Inflammatory inhibition% of MLX–Ethos–OF hydrogel and MLX plain hydrogel.

**Table 1 pharmaceutics-16-00898-t001:** Box–Behnken design expert of MLX-binary ethosomes result.

Run No.	X_1_ Soya Lecithin *w*/*w*%	X_2_ Ethanol *w*/*w*%	X_3_ PG *w*/*w*%	Y_1_ Vesicle Size (nm)	Y_2_ Dispersity Index	Y_3_ EE %	Y_4_ Zeta Potential (mV)
1	2	30	0	174 ± 5.22	0.24 ± 0.024	60 ± 0.02	−39.51 ± 2.90
2	3	30	10	210 ± 6.98	0.17 ± 0.017	82 ± 0.01	−45.5 ± 1.8
3	3	30	10	222 ± 8.17	0.18 ± 0.015	84 ± 0.01	−43.6 ± 0.78
4	3	45	0	337 ± 15.2	0.3 ± 0.011	45 ± 0.02	−42.8 ± 0.34
5	4	15	10	107 ± 3.99	0.31 ± 0.046	50 ± 0.03	−36.3 ± 3.11
6	3	15	0	111 ± 4.42	0.25 ± 0.011	50 ± 0.05	−37.5 ± 1.35
7	2	30	20	269 ± 0.77	0.23 ± 0.018	25 ± 0.04	−45.21 ± 0.94
8	4	45	10	431 ± 29.9	0.47 ± 0.052	14 ± 0.08	−51 ± 0.52
9	3	15	20	141 ± 4.38	0.29 ± 0.010	55 ± 0.04	−47.04 ± 2.87
10	3	30	10	208 ± 9.07	0.2 ± 0.022	88 ± 0.02	−42 ± 1.14
11	2	15	10	121 ± 4.4	0.31 ± 0.038	15 ± 0.03	−48.4 ± 1.04
12	2	45	10	423 ± 17.08	0.41 ± 0.116	36 ± 0.03	−40.1 ± 0.85
13	4	30	20	288 ± 9.29	0.24± 0.014	55 ± 0.04	−46.2 ± 2.68
14	4	30	0	164 ± 6.9	0.25 ± 0.009	43 ± 0.01	−44.8 ± 0.11
15	3	45	20	534 ± 74.55	0.60 ± 0.263	30 ± 0.01	−47.4 ± 4.3
16	3	30	10	221 ± 8	0.22 ± 0.022	86 ± 0.02	−43.5 ± 1.47

**Table 2 pharmaceutics-16-00898-t002:** Response regression analysis of the Box–Behnken design expert.

Y	Model Fit	Adjusted R^2^	Predicted R^2^	R^2^	Mean	SD	C.V	Adequate Precision
VS	Quadratic	0.9975	0.9961	0.9987	247.55	6.29	2.54	95.2284
DI	Quadratic	0.6641	−0.4965	0.7089	0.2908	0.065	22.38	7.9558
EE	Quadratic	0.9803	0.9076	0.9921	51.13	3.40	6.66	27.1098
ZP	2FI	0.8657	0.6807	0.9194	−43.81	1.46	3.34	14.9833

VS: vesicle size, DI: dispersity index, EE% encapsulation efficiency, ZP: zeta potential.

**Table 3 pharmaceutics-16-00898-t003:** ANOVA Affecting Factors for Each Dependent Variable in the Box–Behnken Design.

Parameters	Model *p*-Value	Model Affecting Factors
Vesicle Size nm	<0.0001	Ethanol, propylene glycol
Dispersity Index unit less	0.0180	Ethanol
Encapsulation Efficiency %	<0.0001	Soya lecithin, ethanol, propylene glycol
Zeta Potential Mv	0.0020	Ethanol, propylene glycol

**Table 4 pharmaceutics-16-00898-t004:** MLX–Ethos–OF responses and relative error %.

Response	Predicted Value	Experimental Reading	Relative Error %
Vesicle Size (nm)	157.181	169	7.519357
Dispersity Index	0.175418	0.2087	18.97297
Encapsulation Efficiency %	82.7779	83.1	0.389114
Zeta Potential (mV)	−41.885	−42.76	2.089053

**Table 5 pharmaceutics-16-00898-t005:** The release kinetics model results.

Kinetic Model	MLX–Ethos–OF			
	K	R^2^	AIC	MSC
Zero-order	K_0_ = 1.623	0.9692	10.94	2.81
First order	K_1_ = 0.021	0.9981	2.59	5.59
Higuchi model	K_H_ = 7.724	0.8959	14.59	1.59
Korsmeyer Peppas model	K_KP_ = 3.099, *n* = 0.799	0.9999	−8.38	9.26
Baker-Lonsdale	K_BL_ = 0	0.7696	11.96	0.8

K represents the model constant.

**Table 6 pharmaceutics-16-00898-t006:** pH and spreadability measurements of the hydrogel.

Gelling Agent %	pH Value	Spreadability cm^2^
H.A. 1%	6 ± 0.1	22.7 ± 14.9
H.A. 2%	6.3 ± 0.3	15.5 ± 1.9
Plain MLX 1%	6 ± 0.06	17.6 ± 1.6
Plain MLX 2%	6.1 ± 0.11	11.9 ± 6.5

## Data Availability

The raw data supporting the conclusions of this article will be made available by the authors on request.

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
