# Peer review of "Formulation Development of Meloxicam Binary Ethosomal Hydrogel for Topical Delivery: In Vitro and In Vivo Assessment"

_pharmaceutics, 2024, doi:10.3390/pharmaceutics16070898_

Round 1
Reviewer 1 Report
Comments and Suggestions for Authors
Formulation Development, Optimization by Response Surface 2 Methodology (RSM) Box-Behnken Design Expert of Binary 3 Ethosome-Loaded Meloxicam In vitro and In vivo Hydrogel As- 4 sessment for Topical Anti-inflammatory Response
1) The title is too long!
2) Figure 2 - it is not informative and is not completely clear in terms of designations and meanings
3) equations 8-10- it is better to present with explanations in the supplement
4) It is necessary to highlight the novelty, discuss the advantages and compare the developed systems with similar systems presented in the literature
Comments on the Quality of English LanguageModerate editing of English language required
Author Response
thanks for reviewing the manuscript, and thanks for the comments that aid in enhancing the total work. All comments were answered individually. Also, several improvements were made to the manuscript by rewriting the paragraphs.

Reviewer 2 Report
Comments and Suggestions for Authors
The authors investigate a technique for topical meloxicam delivery using binary ethosomes within a hyaluronic acid hydrogel. While the research presents interesting findings, several areas could benefit from clarification and further expansion.
I have the following suggestions:
1. The title is difficult for understanding from the general reader. One possible revision could be, for example, "Binary Ethosomes as a Promising Vehicle for Topical MLX Delivery: Formulation, Optimization, and In Vivo Evaluation".
2. The authors mention the advantages of binary ethosomes over conventional liposomes and classical ethosomes. However, a more in-depth discussion is needed on how the addition of propylene glycol (PG) to ethanol contributes to better co-solvency, vesicle size reduction, and enhanced permeability. Is there a potential synergistic effect of these two alcohols in the context of MLX encapsulation and delivery?
3. Figure 2 could be moved to a supporting information. It rises a lot of questions and confuses the reader.
4. A More Detailed Analysis of In Vitro Release Data is need, for example the release data were fitted to different kinetic models, but it only mentions the R-squared values for model selection. A more comprehensive analysis of the release kinetics (rate constant K) would provide valuable insights.
5. The authors did not compare the binary ethosomes to other lipid based drug delivery systems, such as liquid crystalline cubosomes or hexosomes to demonstrate their advantages and disadvantages. A recent paper proposed plasmalogen based liposome drug delivery with an outstanding performance; “Lipid and Transcriptional Regulation in a Parkinson’s Disease Mouse Model by Intranasal Vesicular and Hexosomal Plasmalogen-Based Nanomedicines”, Adv. Healthcare Mater. 2024, 13, 2304588
Comments on the Quality of English LanguageSome sentences need to be formulated more clearly.
Author Response
Thanks for your time and effort in reviewing the manuscript. I replied to the comments, and the manuscript underwent several improvements mentioned in the report.

Round 2
Reviewer 1 Report
Comments and Suggestions for Authors
Dear Authors, Thanks for the revision version of the paper! but still there are the points to the improvement of the VS:
1) Abstract – should be rewrite according to the references made in the article and pay more attention to the experimental results obtained in the work: including discussion of : In vitro Drug Release% from MLX-Ethos-OF nanosuspension; Ex vivo and In vivo Anti-inflammatory activity of MLX-Ethos-OF Hydrogel etc
2) Fig. 2 – poor quality and nothing is clear – process it to make it more clear
3) Fig. 6 - it looks more like raw data, usually all this is not shown, but only the main thing, and the table separately
The rest figures could be included in the supplement
4) Table 7 uneven font
Comments on the Quality of English Language
Minor editing of English language required
Author Response
Thanks for reviewing the article. The article undergoes several improvements in the abstract, introduction, design method, results, and conclusion.
